# Self-Powered Resistance-Switching Properties of Pr_0.7_Ca_0.3_MnO_3_ Film Driven by Triboelectric Nanogenerator

**DOI:** 10.3390/nano12132199

**Published:** 2022-06-27

**Authors:** Yanzi Huang, Lingyu Wan, Jiang Jiang, Liuyan Li, Junyi Zhai

**Affiliations:** 1Center on Nano-Energy Research, Guangxi Key Laboratory for Relativistic Astrophysics, School of Physical Science and Technology, Guangxi University, Nanning 530004, China; 1907301031@st.gxu.edu.cn (Y.H.); 2007301061@st.gxu.edu.cn (J.J.); 2007301071@st.gxu.edu.cn (L.L.); 2CAS Center for Excellence in Nanoscience, Beijing Key Laboratory of Micro-Nano Energy and Sensor, Beijing Institute of Nanoenergy and Nanosystems, Chinese Academy of Sciences, Beijing 100083, China; 3College of Nanoscience and Technology, University of Chinese Academy of Sciences, Beijing 100049, China

**Keywords:** self-powered, resistance-switching, triboelectric nanogenerator

## Abstract

As one of the promising non-volatile memories (NVMs), resistive random access memory (RRAM) has attracted extensive attention. Conventional RRAM is deeply dependent on external power to induce resistance-switching, which restricts its applications. In this work, we have developed a self-powered RRAM that consists of a Pr_0.7_Ca_0.3_MnO_3_ (PCMO) film and a triboelectric nanogenerator (TENG). With a traditional power supply, the resistance switch ratio achieves the highest switching ratio reported so far, 9 × 10^7^. By converting the mechanical energy harvested by a TENG into electrical energy to power the PCMO film, we demonstrate self-powered resistance-switching induced by mechanical movement. The prepared PCMO shows excellent performance of resistance switching driven by the TENG, and the resistance switch ratio is up to 2 × 10^5^, which is higher than the ones ever reported. In addition, it can monitor real-time mechanical changes and has a good response to the electrical signals of different waveforms. This self-powered resistance switching can be induced by random movements based on the TENG. It has potential applications in the fields of self-powered sensors and human-machine interaction.

## 1. Introduction

With the advantages of fast switching speed, high storage density, and low power consumption, resistive random access memory (RRAM) has been considered a promising candidate for the next generation of non-volatile memories (NVMs) storage technology [1,2,3,4]. The RRAM cell has a capacitor-like structure composed of insulating or semiconducting materials sandwiched between two metal electrodes. In 1962, Hickmott first reported hysteretic current-voltage (I–V) characteristics in the metal-insulator-metal (MIM) structure of Al/Al_2_O_3_/Al [5], indicating that resistive switching occurs as a result of applied electric fields, i.e., the resistive switching phenomenon. RRAM switches back and forth between a high resistance state (HRS) and a low resistance state (LRS) by the action of an applied electric field. A recent study has shown that the switching speed was faster than 5 ns [6]. In addition, I.G. Baek et al. showed that the metal oxide-based RRAM reached up to 10^12^ read cycles [7], proving that oxide-based RRAM has excellent memory performance. Meanwhile, due to the simplicity of the RRAM structure, the highest-density integrated array could be realized. The RRAM-based intersection structure reported by ShuKai Duan et al. demonstrated the feasibility and effectiveness of this scheme [8,9]. Besides, T. Kawauchi et al. reported the resistive switch-based artificial synaptic devices [10,11,12], indicating the great potential of RRAM for the development of highly networked and neuromorphic circuits. Various studies have demonstrated that RRAM has great prospects in the application of NVMs. Currently, with the intensification of the energy crisis, reducing energy consumption has become a consensus, and developing self-powered devices has become a research front. RRAM driven by traditional external power sources will consume huge energy in large-scale applications. Therefore, self-powered non-volatile memory devices are highly desirable.

Recently, the triboelectric nanogenerator (TENG) has attracted extensive interest as a novel energy collector. It converts mechanical energy into electrical energy based on the coupling between triboelectrification and electrostatic induction [13,14]. With the advantages of simple structure and high output in low frequency and low cost, TENG has been widely used in the development of self-powered sensors. Lots of self-powered sensors have been successfully demonstrated, such as pressure sensors [15], displacement sensors [16], acceleration sensors [17], vibration and acoustic sensors [18,19], multi-function sensors [20,21], and so on. The TENG that works in mechanical motion mode also supplies a new concept to prepare a self-powered RRAM. The TENG collects external mechanical signals and converts them into electrical signals for driving resistive switching, writing, and storing data without an external power supply. It can also monitor in real-time mechanical changes in the environment. Bo-Yun Kim reported a resistive switching memory Integrated with a nanogenerator, the switching ratio of (Na_0.5_K_0.5_)NbO_3_ is greater than two orders of magnitude [22]. Zhengchun Yang designed a self-powered RRAM based on Ga_2_O_3_ with a switching ratio of about 10^2^ [23]. Although very few self-powered RRAMs have been reported, their performance is relatively low. The property of self-powered RRAM is crucial and can be further improved.

In this work, we report a self-powered RRAM and its self-driven resistive switching (RS) performance based on the TENG. We have prepared a sandwiched structure of Pt/PCMO/Ag, in which the PCMO film acts as a resistive switching layer. The TENG harvests the kinetic energy of movement and converts it into electrical energy to induce resistive switching in PCMO films. With a traditional power supply, the resistance switch ratio achieves the highest switching ratio reported so far, 9 × 10^7^. The self-powered RRAM demonstrates a switching ratio of up to 2 × 10^5^ in self-powered mode, which effectively converts external stimuli into electrical signals to realize resistive switching. The performance is 1000 times higher than the existing reported values. Meanwhile, this device exhibits a good response to electrical signals of different waveforms. The constructed device exhibits excellent sensitivity to mechanical movement and records each mechanical change in real-time. The self-powered RRAM shows an excellent response and good stability over multiple switching cycles. Besides the application in non-volatile memory, this self-powered RRAM has potential applications in the fields of self-powered sensors and human-machine interaction.

## 2. Materials and Methods

Preparation of RRAM: First, the Pt/SiO_2_/Si substrate is cleaned with acetone and ethanol, followed by ultrasonic treatment in pure water. The PCMO thin films are deposited on a Pt/SiO_2_/Si substrate by means of radio-frequency (RF) magnetron sputtering. The growth temperature is 383.15 K, and the base pressure of the sputtering chamber is 3 × 10^−4^ Pa. The working pressure is 0.43 Pa and is maintained by a gas mixture of oxygen and argon with a flow of 1 sccm and 24 sccm, respectively. After the fabrication, X-ray diffraction (XRD) and scanning electron microscope (SEM) are used to assess the crystalline quality of PCMO films. In addition, a 30-nm-thick Ag top electrode is deposited on the PCMO by vacuum evaporation with a shadow mask. The resistance switching performances of Pt/PCMO/Ag are measured by Keithley 2612B Source Meter at room temperature.

Fabrication of TENG: The TENG consists of a conductive sponge attached to a polymethyl methacrylate (PMMA) substrate and a Polytetrafluoroethylene (PTFE) membrane with a conductive tape attached to another PMMA substrate. The conductive sponge and the conductive tape are purchased from Guangzhou Beilong Electronics Co., Ltd., Guangzhou, China. The size of the TENG is 4 cm by 4 cm. It can work in both contact-separation and sliding modes. The output voltage and current of TENG are measured by an electrometer (Keithley 6514). The TENG is driven by a mechanical linear motor to work.

## 3. Results and Discussion

Figure 1a shows the schematic of the self-powered RRAM. The RRAM has a vertical sandwiched structure of Pt/PCMO/Ag. The two electrodes of the RRAM are connected with those of the TENG to form a current loop. Meanwhile, the TENG harvests the kinetic energy of motions to induce the resistive switching (RS) of RRAM. 

Firstly, the properties of PCMO thin films are characterized by using the XRD, and the XRD patterns of four samples are shown in Figure 1b. Three main diffraction peaks can be clearly observed at about 33.1°, 47.8°, and 56.4° in the XRD pattern, respectively, corresponding to (121), (202), and (113) diffraction peaks of the face-centered cubic structure of PCMO [24]. In order to measure the thicknesses of PCMO films prepared under different sputtering times, the cross-sectional images of four samples are observed by SEM, as shown in Figure 1c–f. Samples with a sputtering time of 0.5 h, 1 h, 1.5 h, and 2 h are labeled as S1, S2, S3, and S4, respectively. The thickness of the PCMO layer in each sample is designed as 110 nm, 250 nm, 300 nm, and 500 nm, respectively.

Figure 2 shows the RS characteristics of the PCMO memory cells measured through a Keithley 2612B Source Meter with a traditional power supply. The inset shows typical I–V curves of a PCMO memory device. In order to avoid the dielectric breakdown of the device, the limiting current is set to 100 mA during the measurements. Applying a dc voltage cycle from positive to negative (0 V → 2.5 V → 0 V → −2.5 V → 0 V) to the PCMO memory device, four samples exhibit similar I–V features. When a positive dc voltage scan is applied to the PCMO memory devices, samples S1, S3, and S4 are in a low resistance state (LRS), and sample S2 is in a high resistance state (HRS). When a negative dc voltage scan is applied to the PCMO memory device, the samples are in the HRS at first and then produce a SET-process-induced resistance shift from the HRS to the LRS. As the applied reverse bias increases to a certain value (SET voltage), the device switches from an HRS to an LRS. There is an abrupt increase in the current at the SET voltage. The SET voltages of the four samples are −2.2 V, −0.6 V, −2.1 V, and −1.7 V, respectively. The SET voltages of the samples are relatively low, which is conducive to reducing the requirement of the TENG output and making them easier to realize self-powered RRAM. In Figure 2, the I–V curves of the four samples show obvious hysteresis characteristics, and a large hysteresis window indicates that there is a large switching ratio of HRS/LRS, namely, a high resistance-switching ratio. The switching ratio of the four samples is 9 × 10^7^, 2.5 × 10^3^, 1.2 × 10^4,^ and 1 × 10^6^, respectively. Among them, sample S1 achieves the highest switching ratio reported so far.

Retention performance is an important factor in evaluating the RS behavior for an RRAM device. Thus, we measured the retention performance of the device in the HRS and the LRS at a reading voltage of 0.1 V. As shown in Figure 2, the PCMO memory unit had excellent retention characteristics during the 10^3^ s testing. In the absence of other electrical stimulation, the device will remain in its original state. No significant change is observed in the HRS and LRS; the resistance states of the resistors are also not reversed. The results indicate that the Ag/PCMO/Pt resistive storage structure has good retention characteristics. Furthermore, we investigate self-powered resistance properties driven by the TENG.

To evaluate the feasibility of TENG inducing the RS, its electrical output properties are measured. Among the four working modes of TENG, the vertical contact-separation mode and the sliding mode can generate higher output. The TENG working in these two modes is chosen to apply in four RRAM samples. The TENG is operated by a mechanical linear motor to contact and separate periodically, and its electrical outputs are measured using an electrometer (Keithley 6514). The typical open-circuit voltage (V_oc_) and short-circuit current (I_sc_) of TENG working in the vertical contact-separation mode are shown in Figure 3, which are alternating pulse outputs. As the movement amplitudes are 10 mm, 20 mm, 30 mm, and 40 mm, the peak values of V_oc_ are 15.2 V, 16.1 V, 18.0 V, and 21.1 V, respectively, as shown in Figure 3a. The peak values of I_sc_ are 380 nA, 403 nA, 458 nA, and 469 nA, as shown in Figure 3b. The outputs of V_oc_ and I_sc_ of the TENG are increased with the variation of amplitudes.

Figure 3c,d shows the V_oc_ and I_sc_ of TENG working in lateral sliding mode. When the top layer slides away about 5 mm, there is a 1.17 V potential difference between the two electrodes of the TENG. With an increase in the sliding amplitudes from 5 mm to 20 mm, the V_oc_ is gradually increased from 1.17 V to 13.4 V, as shown in Figure 3c. In addition, the I_sc_ of the TENG are 11 nA, 20.4 nA, 35.7 nA, and 40.5 nA, as shown in Figure 3d. The outputs of the TENG increase with the sliding amplitudes.

It is noted that whether in the vertical contact-separation mode or the lateral sliding mode, the electrical outputs of the TENG are alternating cyclic signals with a change of 0 V → positive peak Voltage → 0 V → negative peak voltage → 0 V, which are exactly the required stimulation signal of RS. Such a pulse alternating electric signal is very suitable for the application of RS. Due to the sensitive response to external forces, the RRAM with TENG has great potential in the application of self-powered sensing and human-machine interaction. Moreover, TENG is a high efficient energy harvester in low frequency by converting mechanical energy to electricity. It is an excellent candidate for developing a self-powered RRAM device.

By utilizing a TENG working in the vertical contact-separation mode to our four PCMO samples, as shown in Figure 4a, the RS properties are measured. With an amplitude of 13 mm of linear motor motion, the TENG successfully induces resistance switching in four PCMO films. It is observed that the RS of PCMO synchronously changes with the movement state of the TENG. While the TENG is being separated and outputs a positive electrical pulse, the PCMO exhibits the initial HRS. With the TENG moving from the separation state to the contact state, the positive electrical pulse becomes a negative pulse, and the RS of PCMO correspondingly shifts from the HRS to the LRS. In turn, as the TENG moves from the contact state to the separation state, the PCMO returns to the HRS. The RS of PCMO successfully switches with the movement state of TENG. The switching ratios of four samples, S1, S2, S3, and S4, are 2 × 10^5^, 588, 9.6 × 10^3^, and 1.8 × 10^5^, respectively. Compared with the switching ratios induced by the electric field powered by a Source Meter, sample S4 shows the same switching ratio, while the switching ratios of samples S1, S2, and S3 are slightly lower. The HRS of sample S1 is more affected by TENG transient output in the switching resistance state. Samples S2, S3, and S4 are relatively stable in HRS.

As the output of the TENG in the sliding mode is lower than that in the contact-separation mode, only samples S2 and S4 with lower set voltages are observed. The induced resistance switching effect, when the sliding amplitude of the TENG is 30 mm, is shown in Figure 4b. While the upper friction layer slides outward, a positive peak voltage of 28 V is generated, and the resistance value of the PCMO decreases and switches to the LRS. When the upper friction layer slides inward, a reversely increasing voltage is generated to make the PCMO return to the HRS. The switching ratios of the samples S2 and S4 are in the order of 10 and 2.4 × 10^2^, respectively. The switching ratios are the same as those measured by a Source Meter at that time.

Figure 4c shows the stability of the switching ratio test over 50-cycles for the device as the TENG is working in contact-separation mode. During the test, the switching ratios of S2 and S3 slightly decrease. The switching ratios of samples S2 and S3 are about 844 and 1.8 × 10^4^ at the beginning. Then they decrease slightly and stabilize at about 500 and 1 × 10^4^. The switching ratio of the sample S1 fluctuates between 3 × 10^5^ and 1.6 × 10^5^. Sample S4 is similar to S1. All the samples demonstrate good stability in the retention test, which means the self-powered RRAM devices all have the perfect capability to store data and keep the data quite stable. This approach shows good reliability since it can ensure the correct switching of the device in each cycle.

Figure 4d shows the stability of the switching ratio test over 50-cycles for the device as the TENG is working in the sliding mode, exhibiting a slight change in the switching ratio. In this mode, the pulse signal with narrow width can also excite RS, while the HRS and LRS fluctuate within a certain range due to the fluctuation of TENG output. The TENG working in the contact-separation mode is more likely to stably drive RS, while the TENG working in the sliding-friction mode is not easy to drive the stable RS phenomenon due to the asymmetric output voltage.

As shown in Figure 5, the possible mechanism is a summary to explain the effect of TENG on the RS behaviors. In p-type oxide semiconductors, it is generally considered that the migration of oxygen vacancies induces resistance switching [25,26]. In the contact-separation mode, when the conductive foam and PTFE approach, the negative charges on the PTFE films induce the positive charges in the conductive tape, which produces the current from the conductive tape to the conductive foam through RRAM, leading to the migration of positively charged oxygen vacancy and forming conductive filaments. Therefore, the PCMO switches from the HRS to the LRS. As the conductive foam and the PTFE are being separated, the negative charges on the PTFE film appeal to a number of positive charges in the conductive tape, which produces a current flowing from the conductive foam to the conductive tape through RRAM. The filaments are ruptured, and the PCMO is back to the HRS.

In sliding mode, when the conductive foam is slid backward, the redundant transferred charges on the electrodes will flow from the conductive tape to the conductive foam with an increase in the contact area. Therefore, the PCMO switches from the HRS to the LRS. When the conductive foam starts to slide outward, triboelectric charges are not fully compensated at the mismatched areas, resulting in the potential difference across the two electrodes, which produces a current flowing from the conductive foam to the conductive tape and forms conductive filaments. The filaments are ruptured, and the PCMO is back to the HRS.

## 4. Conclusions

In summary, we demonstrate a self-powered RRAM based on the PCMO film and the TENG. The TENG collects external mechanical energy and converts it into electrical energy to provide an electric field in PCMO, and the electric field successfully induces the RS effect. The prepared PCMO exhibits excellent self-powered RS performances, and the largest switching ratio is up to 2 × 10^5^, which is the highest self-powered performance of RRAM reported at present. The self-powered RRAM effectively responds to external mechanical motions to achieve resistance switching. The constructed device exhibits excellent sensitivity to mechanical movement and records each mechanical change in real-time. We find that the TENG working in the contact-separation mode is more likely to drive the RS effect. In addition, the switching ratios maintain good stability in multiple switching cycles. The results are of significance for applications of RRAM in fields such as self-powered sensors and human-machine interaction. All authors have read and agreed to the published version of the manuscript.

## Figures and Tables

**Figure 1 nanomaterials-12-02199-f001:**
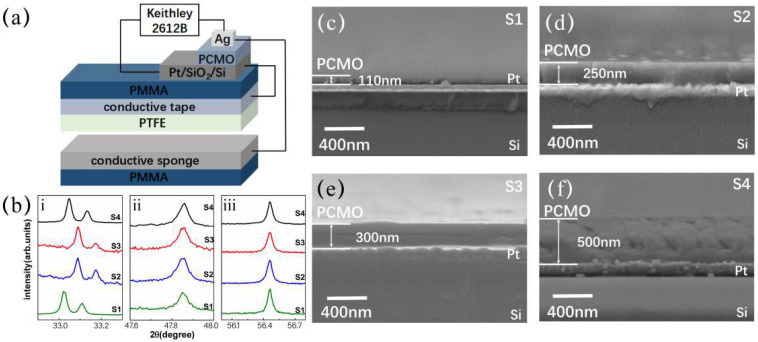
(**a**) Schematic illustration of self-powered RRAM; (**b**) XRD patterns of PCMO films deposited on Pt/SiO_2_/Si substrates at room temperature, and three main diffraction peaks are shown in (i), (ii), and (iii); (**c**–**f**) Cross-sectional SEM images of the four samples with PCMO of different thicknesses, (**c**) 110 nm, (**d**) 250 nm, (**e**) 300 nm.

**Figure 2 nanomaterials-12-02199-f002:**
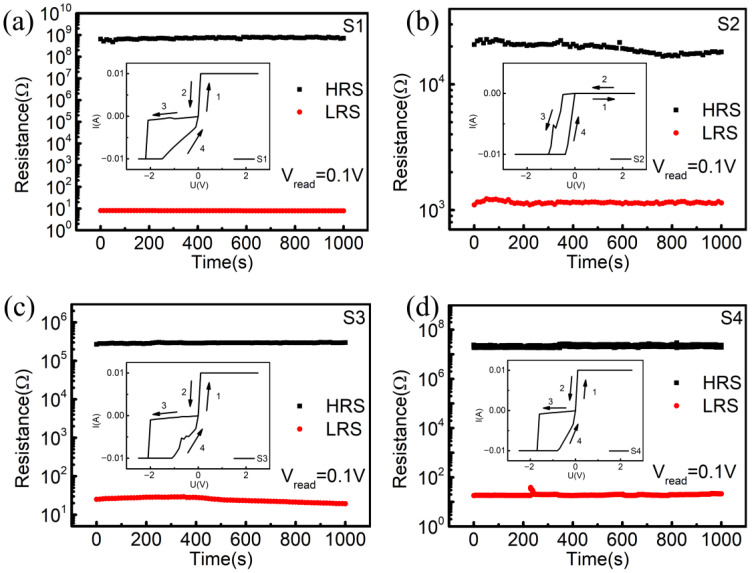
The resistive switching characteristics of (**a**) S1, (**b**) S2, (**c**) S3 and (**d**) S4. The insets show the I–V curves of each sample.

**Figure 3 nanomaterials-12-02199-f003:**
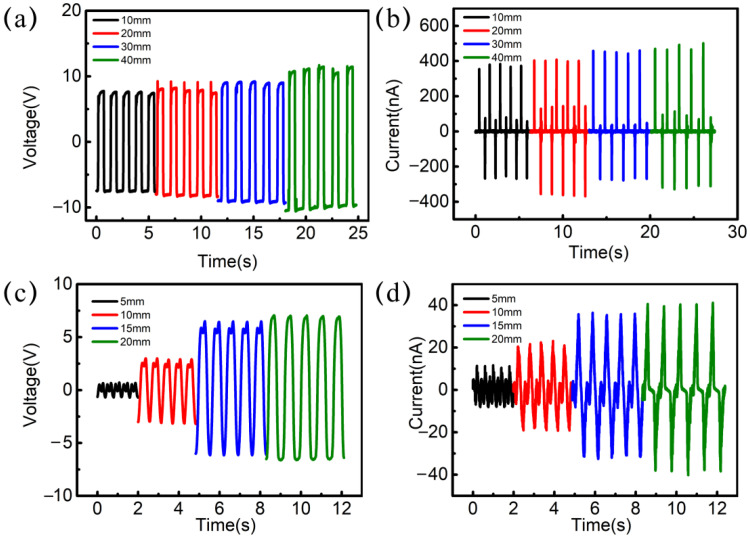
Output performance of the TENG. (**a**) Open-circuit voltage and (**b**) short-circuit current with a cyclic contact force of different amplitudes based on contact mode. (**c**) Open-circuit voltage and (**d**) short-circuit current of the TENG based on lateral sliding mode.

**Figure 4 nanomaterials-12-02199-f004:**
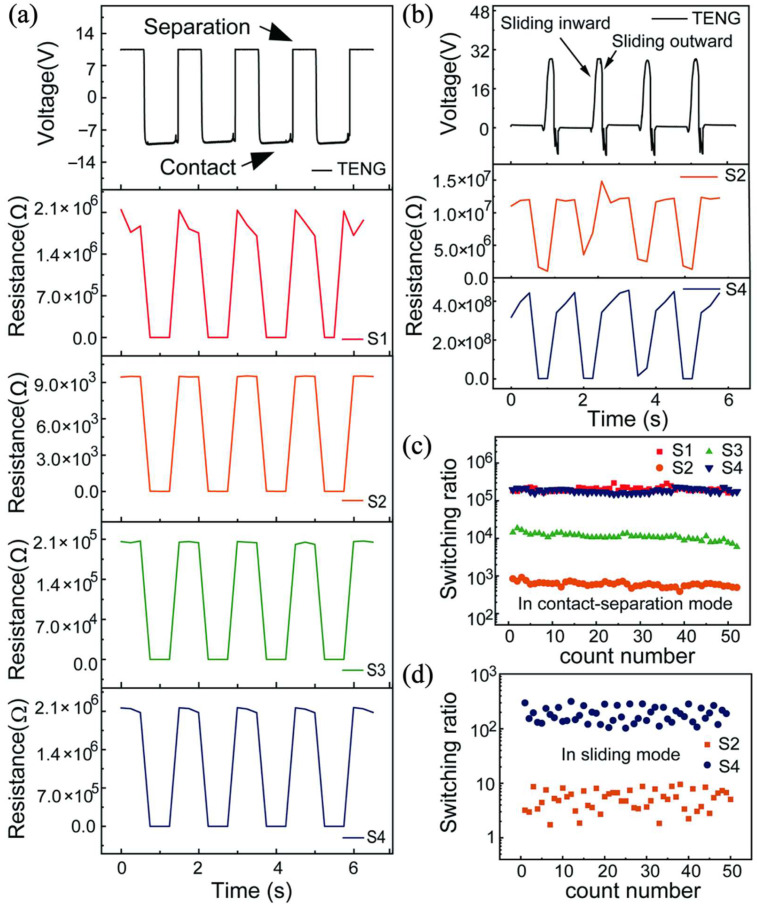
RS in an Ag/PCMO/Pt cell at room temperature. By applying voltages from TENG, the resistance of the cell changes reversibly between HRS and LRS. (**a**) RS characteristics of four samples based on contact mode TENG. (**b**) RS characteristics curves of samples S2 and S4 based on sliding mode TENG. (**c**) Endurance of switching behavior in the contact-separation mode. (**d**) Endurance of switching behavior in the sliding friction mode.

**Figure 5 nanomaterials-12-02199-f005:**
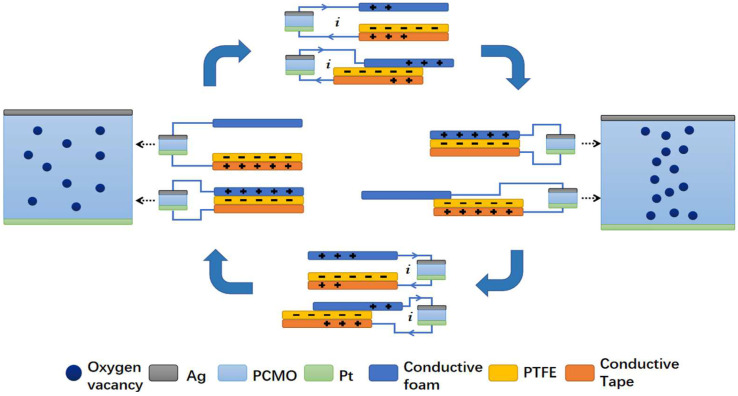
Schematic illustration of the underlying mechanism of the self-powered RRAM.

## Data Availability

Not applicable.

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
