# Peer review of "Self-Powered Resistance-Switching Properties of Pr0.7Ca0.3MnO3 Film Driven by Triboelectric Nanogenerator"

_nanomaterials, 2022, doi:10.3390/nano12132199_

Round 1

Reviewer 1 Report

In this work the authors study triggering an RRAM using TENG. They use resistive switching Pr0. 7Ca0. 3MnO3 for the RRAM.

While the study is well conducted, a clear application is not shown by the authors. The work can be published after the following comments are addressed.

  1. Is the switching ratio = HRS/LRS? This should be stated in the manuscript.
  2. What material is the "conductive tape" and "conductive sponge". Specify material supplier information if it is not standard material.
  3. Line 136 states S1, S2, S3, S4 switching ratios of 9 × 1E7, 2.5 × 1E3, 1.2 × 1E4 and 2 × 1E4. For TENG powered, line 190 states S1, S2, S3, S4 switching ratios of 2×1E5, 588, 9.6×1E3, and 1.8×1E5. Why is the S4 switching ratio higher while S1,S2,S3 lower? TENG has higher voltage than the power supply (2.5V). Does it make any difference?
  4. Figure 4 caption states R-T curves. What is R-T full form? Why Figure 4d has dual y-axis when the value seems close to zero for both S2 and S4?
  5. The mechanism explained in the last two paragraphs seems incorrect. During PTFE approaching sponge, positive charge will flow from conductive tape through RRAM and end up on sponge surface. During PTFE separating sponge, positive charge will flow from sponge through RRAM and end up on conductive tape. A schematic figure should be drawn to clearly explain this.

Author Response

Response 1: Thank you so much for your good suggestion. Yes, the switching ratio is HRS/LRS, and we have stated it in the revised manuscript. (Page 4 Line 140)

Response 2: We thank the reviewer for detailed review. The photographs of conductive tape and conductive sponge are shown in Figure S1. The conductive tape is a type of conductive cloth plated with metal nickel and copper. The conductive sponge is a type of polyurethane sponge plated with metal nickel and copper. The material supplier is Guangzhou Beilong Electronics Co., Ltd. We have added this information in Page 3 Line 99-101.

Response 3: Thank you for your careful review and good questions. We rechecked our original data of sample S4. It is found that we used a wrong data to calculate the switching ratio of sample S4 when it is powered by a traditional power source. The right switching ratio of S4 is 1E6 and we have corrected it in the revised manuscript (Page 4 Line 142). Due to the TENG outputs a pulse electrical signal with higher peak voltage but lower average power, the switching ratios of four samples become lower when they are driven by a TENG.

    The power supply and the TENG have different electrical output characteristics. As shown in Figure S2, in a RS switching process, the power supply outputs a voltage cycle from positive to negative (0 V → 2.5 V → 0 V → -2.5 V → 0 V) with a step of 0.1 V at a frequency of 50 Hz. The TENG outputs pulse voltage with a certain duty cycle. Its peak voltage is high but the average voltage is lower. The average power of TENG is lower than that of power supply.

Response 4: Thank you for your careful review. The complete form of R-T is Resistance – Time. In Figure 4, the curves indicated that the resistance of RRAM varies with the output of TENG. We have replaced R-T with Resistive switching (RS) characteristics in Figure 4 caption (Page 6 Line 213-214). In Figure 4d, there's really no need for using dual y-axis and we have revised it to a single y-axis.

Response 5: Thank you for pointing out this incorrect description. In some sentences, our descriptions for the conductive tape and the conductive sponge are opposite. Following the reviewer’s suggestion, we have revised them in the manuscript (Page 7 Line 238-253). We have also added a schematic figure to explain the mechanism more clearly (Page 7 Line 234-235).

Special thanks to you for your good comments.

Reviewer 2 Report

An interesting paper that demonstrates the benefits of TENGs as autonomous power supplies.

I think the paper would be improved if the authors would make clearer that TENGs based on contact electrification (involving electrical insulators such as in this study) produce decent voltages but extremely low current densities. It should be made clearer that the current output of such TENGS is an AC signal, meaning that it wold be of limited general scope to power up electronics which by norm require a DC input. The authors should discuss this in the introduction and conclusions, and alert the readership of the possible DC alternatives. There are now several designs of DC TENGs, the most notable being that of silicon-based sliding Schottky contacts, such as recently demonstrated in Nano Energy 2022 93:106861. The impact of the paper would be improved by stressing further on this.

Author Response

Response to Reviewer 2 Comments

Response: Thank you for your good comments and suggestion. In RRAM applications, the polarity of the applied voltage is changed to achieve the RS switching (set and reset process). One electrical field induces a transition from a HRS to LRS, and an opposite electrical field is needed to switch the RRAM cell back into the HRS. The TENGs generated AC signals through reciprocating mechanical motions are suitable for a complete RS switching cycle of RRAM. As for a DC-TENG, we need to change its polarity in the RS switching process. We will consider how to utilize a DC-TENG to a RRAM device in future.

Special thanks to you for your good comments.
